# Impact of COVID-19 Pandemic on Adherence to Chronic Therapies: A Systematic Review

**DOI:** 10.3390/ijerph20053825

**Published:** 2023-02-21

**Authors:** Elena Olmastroni, Federica Galimberti, Elena Tragni, Alberico L. Catapano, Manuela Casula

**Affiliations:** 1Epidemiology and Preventive Pharmacology Service (SEFAP), Department of Pharmacological and Biomolecular Sciences, University of Milan, 20133 Milan, Italy; 2IRCCS MultiMedica, 20099 Sesto San Giovanni (MI), Italy

**Keywords:** COVID-19 pandemic, medication adherence, chronic therapies, healthcare system, epidemiology

## Abstract

The spread of the coronavirus disease 2019 (COVID-19) pandemic caused a sudden and significant disruption in healthcare services, especially for patients suffering from chronic diseases. We aimed at evaluating the impact of the pandemic on adherence to chronic therapies through a systematic review of available studies. PubMed, EMBASE, and Web of Science were searched since inception to June 2022. Inclusion criteria were: (1) observational studies or surveys; (2) studies on patients with chronic diseases; (3) reporting the effects of COVID-19 pandemic on adherence to chronic pharmacological treatment, as a comparison of adherence during the pandemic period vs. pre-pandemic period (primary outcome) or as rate of treatment discontinuation/delay specifically due to factors linked to COVID-19 (secondary outcome). Findings from 12 (primary outcome) and 24 (secondary outcome) studies showed that many chronic treatments were interrupted or affected by a reduced adherence in the pandemic period, and that fear of infection, difficulty in reaching physicians or healthcare facilities, and unavailability of medication were often reported as reasons for discontinuation or modification of chronic therapies. For other therapies where the patient was not required to attend the clinic, continuity of treatment was sometimes ensured through the use of telemedicine, and the adherence was guaranteed with drug stockpiling. While the effects of the possible worsening of chronic disease management need to be monitored over time, positive strategies should be acknowledged, such as the implementation of e-health tools and the expanded role of community pharmacists, and may play an important role in preserving continuity of care for people with chronic diseases.

## 1. Introduction

The coronavirus disease 2019 (COVID-19), caused by severe acute respiratory syndrome coronavirus 2 (SARS-CoV-2), has posed major challenges to healthcare systems, mainly during the years 2020 and 2021. Although many countries are currently going through a transitional period, in which health systems are, at different rates, returning to pre-pandemic levels, the effects of COVID-19 pandemic were initially extremely impactful. Beyond the direct impact on morbidity and mortality, the pandemic has determined a sudden and significant disruption in healthcare services, especially for chronic patients [1,2,3].

Particularly for these patients, the continuity of medication therapy is a cornerstone for the effective management of their disease and for avoiding complications [4,5]. Medication adherence is defined as the extent to which a patient’s behavior corresponds with the prescribed medication regime, including time, dosing, and interval of medication intake [6,7]. Non-adherence has been widely reported for many chronic therapies [8].

Adherence is a multifactorial phenomenon that can be influenced by various factors, which are usually attributed to five different dimensions: social and economic factors, therapy-related factors, disease-related factors, patient-related factors, and healthcare system-related factors [9]. Treatment-related factors include the complexity of the treatment regimen and the difficulty of administration, as well as the risk of drug-related adverse events. Factors related to the organization of the health system include the cost of therapies, as well as the accessibility of medicines, facilities, and health personnel. Some of these conditions may have an influence on the so-called intentional non-adherence, namely the conscious decision not to take the medication [10].

The health emergency due to the COVID-19 outbreak has strongly affected some of these factors [11]. Many chronic patients have experienced a gap in their care, and the unavailability of clinicians and other healthcare professionals, along with isolation measures and disruptions in communication, activities, and services during the pandemic, may have resulted in less timely and/or less appropriate clinical care and oversight [12]. This certainly had a major impact on chronically ill patients, whose management is closely dependent on the frequency of clinical visits and continuity of drug therapy. However, the consequences on therapeutic continuity may have been very wide-ranging, primarily because the incidence of SARS-CoV-2 infections, as well as hospitalization and mortality rates, have been quite different across geographical settings. Furthermore, healthcare systems have reacted differently, implementing systems to ensure the continuity of care with various timing and modalities [13].

This systematic review therefore aimed at gathering the evidence available to date on the impact of the pandemic on adherence to chronic therapies, describing the differences in a variety of settings and for different diseases, and discussing the main barriers to adherence that the pandemic raised.

## 2. Materials and Methods

This review is reported according to the Preferred Reporting Items for Systematic Reviews and Meta-Analyses guidelines (PRISMA) [14].

We performed a systematic literature search in MEDLINE (via PubMed), Embase, and Web of Science for articles published until June 2022. In addition to the electronic searches, we crosschecked the references of all included articles. Searching strategies, based on combinations of key terms related to medication adherence and COVID-19 pandemic, are reported in the Appendix A.

We selected eligible articles according to the following predefined inclusion criteria: (i) observational studies and surveys; (ii) on children or adult patients with chronic diseases; (iii) reporting the effects of COVID-19 pandemic on adherence to chronic pharmacological treatment, as a comparison of adherence during the pandemic period vs. pre-pandemic period (primary outcome) or as rate of treatment discontinuation/delay specifically due to factors linked to COVID-19 (secondary outcome).

The choice of two different outcomes depended on the different methodological approaches applied to analyze them, being the first mainly derived from pre-post comparisons and the second from cross-sectional evaluations, as well as surveys, and questionnaires.

Only papers written in English were included. Articles that reported interventions to improve adherence, predicted adherence from model analysis, or surveys evaluating only barriers to adherence without any quantification were excluded.

The study selection (title/abstract screening and full-text screening) was independently performed by two reviewers. Any differences between the reviewers were discussed until a consensus was reached.

All data were extracted using a pre-specified extraction form. Data were extracted by one reviewer, and completeness and accuracy were verified by a second reviewer. For each article, we extracted the following characteristics: first author, year of publication, country, condition/medication, research method, number of subjects involved, study period, and main results. Any disagreements were discussed until consensus.

The methodological quality of observational studies included in the primary outcome evaluation was assessed using the National Institutes of Health (NIH) Study Quality Assessment Tools (https://www.nhlbi.nih.gov/health-topics/study-quality-assessment-tools, accessed on 1 November 2022) for Observational Cohorts and Cross-Sectional Studies. Each assessment question was rated with “yes”, “no”, “unclear” or “not applicable”. The instrument was applied independently by two reviewers. Divergent opinions were discussed among authors and a consensus was reached.

## 3. Results

The search identified 12 studies for the primary outcome and 24 studies for the secondary outcome. The results of the search strategy are also illustrated in the Figure 1.

Among the 12 studies included for the primary outcome evaluation (Table 1), 5 studies were conducted in Europe, 5 were from US/Canada, one from Japan, and one from Uganda. The evaluated therapies were mainly for respiratory disease (3 studies) or for inflammatory disease (3 studies); in 2 cases (subcutaneous denosumab for patients with osteoporosis and infusible biologicals in patients with inflammatory bowel disease), the treatment required the patient to come to the clinic in person. Regarding the study design, there were one survey, 2 time series analyses, and 9 retrospective cohort studies with different data sources (administrative health databases, electronic medication monitors data, medical records). Adherence was evaluated as proportion of coverage (as proportion of days covered [PDC], medication possession ratio [MPR], or proportion of administrations compared to what was recommended; 7 studies) or as rate of discontinuation or missed scheduled injection (4 studies); in one study, the primary adherence (patient properly fills the first prescription for a new medication) was evaluated. Quality assessment of included studies is reported in Appendix A.

In 7 studies, there was a worsening of adherence to therapy in the pandemic period compared to a control period in the previous years while 5 studies found no change or even an improvement of adherence levels during COVID-19 period.

Kaye et al. [15] analyzed adherence to medication in US patients with asthma and chronic obstructive pulmonary disease (COPD), observing a 14.5% increase (53.7% to 61.5%) in mean daily controller medication from the first week of January 2020 to the last week of March 2020. The evaluation of prescription trends for inhaled corticosteroids in asthmatic patients by Dhruve et al. [16] in UK showed a sharp increase in March 2020, representing a 49.9% increase compared with February 2020. They reported a median levels of adherence (MPR) of 54.8% (27.4–95.9%) in 2019 and 54.8% (27.4–106.8%) in 2020 (significant increase, *p* < 0.001), with a decline in adherence in about 20% of patients, as a consequence of difficulty in obtaining a new prescription or concerns about the immunosuppressive properties of inhaled corticosteroids. Conversely, the retrospective cohort analysis of Medicare-enrolled older patients with asthma by Ramey et al. [17] showed that mean adherence (PDC) for all controller medications ranged 75–90% in 2019, with a significantly decrease (*p* < 0.001) to 51–70% in 2020. Lower adherence was associated with low disease severity and with having filled less than 90-day supply for a controller medication.

Studies evaluating therapies that required a specialist visit for their prescription or administration consistently reported a decline in adherence, expressed as missed scheduled appointment. The study of Kahn et al. [18] on US national Veterans Affairs healthcare system found that the proportion of patients with inflammatory bowel disease receiving an infusion within 10 weeks of the prior infusion to infusible biologic (infliximab, inflectra, renflexis, and vedolizumab) decreased from 84.6% in 2019 to 73.6% in 2020 (*p* < 0.0001), with a persistent drop in the weekly number of infusions since late March 2020. De Vincentis et al. [19] assessed adherence to denosumab (as a single 60 mg subcutaneous injection every 6 months) in a cohort of Italian osteoporotic patients, showing a reduction from 96,7% in the pre-COVID-19 period to 87.0% during the lockdown (*p* < 0.0001). They also reported that the majority of patients who were non-adherent and/or discontinued denosumab during the COVID-19 lockdown returned for regular follow-up once pandemic restrictions ceased and that the main reason of non-adherence was that patients were afraid of coming to the hospital due to the COVID-19 contagion risk.

Hasseli et al. [20] investigated the adherence of patients with inflammatory rheumatic diseases to their immunomodulatory medication during the three-month lockdown in Germany. Termination of therapy was reported by only 3% of the patients, without relevant changes compared to what reported before the national lockdown, with results that were independent from the type of rheumatic diseases, the immunomodulatory therapy, and the age of patients. In the study by Uchida et al. [21], assessing discontinuation of biologics in patients with psoriasis, defined as ceasing biologic treatment and never receiving any biologic treatment at least until July 2021, 2.8% of patients discontinued biologic treatment in 2020, compared to 0.6% in 2019.

The only included study conducted in Africa (Uganda), by Wagner et al. [22], found no statistically significant change in electronically measured adherence to antiretroviral therapy in 324 HIV patients, although clinic visits decreased by more than 50% after a national lockdown started, and the risk of patients running out of treatment increased from 5% before the lockdown to 25% three months later.

In the evaluation of adherence to ocular hypotensive medication in US patients with glaucoma conducted by Racette et al. [23], a decline in adherence was observed after the declaration of the pandemic, with a decrease in mean adherence (measured using Medication Event Monitoring System caps) from 83.6% before the pandemic to 68% one year later. Moreover, in some patients, despite stable levels of adherence, a reduced regularity in the timing of eye drop instillations in the periods before and after the onset of the pandemic was described.

In Italy, Romagnoli et al. [24] selected 12,030 hypoglycemic treatment-naïve patients and showed that 6-month adherence (as PDC) was 0.80 in 2019 and 0.79 in 2020; similarly, on 19,699 statin-naïve patients, 6-month adherence was 0.90 in 2019 and 0.92 in 2020. Instead, persistence appeared to be more affected by pandemic: 6-month persistence dropped from 90% in 2019 to 56% in 2020 for patients on oral hypoglycemic drugs, and from 83% to 43%, respectively, for patients on statin therapy.

The impact of the COVID-19 pandemic on discontinuation rate for opioid agonist therapy was evaluated by Garg et al. [25] among Ontario (Canada) residents; no significant changes were observed during the first eight months of the pandemic, neither among those stabilized on therapy nor among those who had more recently initiated treatment.

Finally, Villalobos Violan et al. [26] evaluated primary adherence to allergen immunotherapy in an Allergology Unit of a Spanish hospital, reporting a percentage of treatment initiation of 88.1% during pandemic, compared to 94.6% in the non-pandemic period (*p* = 0.022). Personal decision, economic/labor reasons, and problems with access to the healthcare system were the main reasons for not starting therapy.

**Table 1 ijerph-20-03825-t001:** Summary of included studies (n = 12) comparing adherence to chronic treatments during pandemic period vs. pre-pandemic period (primary outcome).

Ref	Year of Publication	First Author	Country	Patients and Treatment	Study Design	Number of Subjects	Period of Analysis	Measure of Adherence	Results
[18]	2020	Kahn N	United States	Infusible biologics in patients with inflammatory bowel disease	Retrospective study on administrative health databases	5026	January–March 2020 vs. January–March 2019	Missed scheduled injection on time, i.e., at 10 weeks after the date of the previous injection	Adherence was 84.6% in 2019 and 73.6% in 2020 (*p* < 0.0001 for the difference).
[15]	2020	Kaye L	United States	Controller inhaler use in patients with asthma and COPD	Retrospective study on electronic medication monitors data	7578	January–March 2020	Number of actuations divided by the number prescribed, weekly	From the first 7 days of January 2020 to the last 7 days of March 2020, there was a 14.5% increase (53.7% to 61.5%) in mean daily controller medication adherence
[20]	2021	Hasseli R	Germany	Immunomodulatory medications in patients with inflammatory rheumatic diseases	Survey	4252	March–June 2020	Rate of discontinuation	4% of the patients reported to discontinue their medication before the national lockdown; during and after the national lockdown the number of reported discontinuations even decreased
[22]	2021	Wagner Z	Uganda	Antiretroviral therapy in HIV patients	Retrospective study on administrative health databases	324	March 2018–September 2020	Percentage of doses taken as for MEMS caps over the doses prescribed	There was no change in adherence after the lockdown started or at any point during the pandemic.
[19]	2022	De Vincentis S	Italy	Denosumab in patients with osteoporosis	Retrospective study on medical records	501	March 2019–March 2020 vs. March 2020–March 2021	Missed scheduled injection on time, i.e., at 6 months after the date of the previous injection	In the pre-COVID-19 period, 3.3% were found to be non-adherent, compared to 13.0% in the lockdown period
[16]	2022	Dhruve H	UK	Inhaled corticosteroids in patients with asthma	Retrospective study on prescription records	1132	2019 vs. 2020	Medication possession ratio	Median levels of ICS adherence were 54.8% (27.4–95.9%) in 2019 and 54.8% (27.4–106.8%) in 2020 (*p* < 0.001).
[21]	2022	Uchida H	Japan	Biologics in patients with psoriasis	Retrospective study on medical records	15,062	January 2016–December 2020	Rate of discontinuation	2.8% of patients discontinued biologic treatment in 2020, compared to 0.6% in 2019
[25]	2022	Garg R	Canada	Opioid agonist therapy in patients with opioid use disorder	Time series analysis on administrative health databases	80,799	April 2019–November 2020	Rate of discontinuation	No significant step change in the weekly percentage of Ontarians who discontinued opioid agonist therapy following the declaration of the state of emergency
[23]	2022	Racette L	United States	Ocular hypotensive medication in patients with primary open-angle glaucoma	Time series analysis from National Institutes of Health-funded study data	79	March–August 2020	Percentage of doses taken as for MEMS caps over the doses prescribed	Overall mean adherence decreased from 83.6% before the pandemic to 68% 1 year later
[17]	2022	Ramey OL	United States	Asthma controller medications in older adults with asthma	Retrospective study on medical records	1637	January–July 2019 vs. January–July 2020	Proportion of days covered	Adherence significantly decreased (*p* < 0.001) from 55–90% to 51–70% for all controller medications
[24]	2022	Romagnoli A	Italy	Hypoglycaemic drugs and statins	Retrospective study on administrative health databases	31,729	January 2011–December 2020	Proportion of days covered	Adherence data ranged from values of 0.79 and 0.75 in 2012 to 0.92 and 0.79 in 2020 for the hypoglycaemic group and statin group, respectively. Persistence curves stratified by year showed a statistically significant difference for both groups under analysis (*p* < 0.0001).
[26]	2022	Villalobos Violán V	Spain	Allergen immunotherapy	Retrospective study	446	March–September 2020 vs. March–September 2019	Primary adherence (first prescription filled)	The percentage of adherence (treatment initiation) in the non-pandemic period was 94.6% and 88.1% in the pandemic period (*p* = 0.022)

Out of the 24 studies included for the secondary outcome evaluation (Table 2), 8 were conducted in Europe, 8 in the Middle East, 3 in US/Canada, 3 in Asia, 1 in Mexico, and 1 in Australia. Evaluated therapies were mainly drugs for inflammatory disease or for transplant recipients (17 studies). Studies were mostly web-based or telephonic surveys (20 studies).

In 3 studies [27,28,29] on immunosuppressive therapies, no patients reported to have discontinued their treatment due to COVID-19 concerns. In 6 studies [30,31,32,33,34,35], the rate of discontinuation for reasons associated with pandemic was less than 5%, and in 8 studies [36,37,38,39,40,41,42,43] was between 5% and 10%.

In France, Constantino et al. [44] conducted a survey on the adherence to medications for chronic inflammatory rheumatic disease, finding that more than 30% of patients suspended or decreased the dosage of one of their drugs during COVID-19 pandemic, with 25.2% of subjects reporting a treatment modification for fear of infection. In the study by Kulhas Celik et al. [45] evaluating the effect of patient and parental anxiety on adherence to subcutaneous allergen immunotherapy administered in a Turkish pediatric allergy and immunology hospital clinic during COVID-19 pandemic, 20.5% cited fear of COVID-19 transmission as primary reason of non-adherence.

In the survey conducted by Oguz Topal et al. in Turkey [46], patients with psoriasis were enrolled and asked to report any reduction of the dosage of the medication, treatment interruption, or temporary suspension. Of 342 patients, 45.9% either discontinued medications or reduced the dosage, mainly because they were unable to go to the hospital (19.2%) or they had concern about the COVID-19 infection (16.3%). In the study on adherence to antiglaucoma eyedrops conducted by Subathra et al. [47] in India, 31.4% of patients reported treatment discontinuation because medicines were not available. Akour et al. [48] interviewed 431 individuals who suffer from chronic diseases in Jordan via a web-based questionnaire, and found that 22.7% of patients stopped or decreased medication intake during pandemic period due to the impossibility to access drugs at clinics.

**Table 2 ijerph-20-03825-t002:** Summary of included studies (n = 24) reporting the rate of treatment discontinuation/delay specifically due to factors linked to COVID-19 (secondary outcome).

Ref	Year of Publication	First Author	Country	Patients and Treatment	Methods	Number of Subjects	Period of Analysis	Results
[27]	2020	Georgakopoulos JR	Canada	Apremilast in patients with psoriasis	Patient Support Program	188	February–April 2020	No patients had discontinued treatment due to COVID-19 concerns
[31]	2020	Georgakopoulos JR	Canada	Dupilumab in patients with atopic dermatitis	Patient Support Program	162	February–April 2020	1 patient (0.62%) had temporarily discontinued treatment due to COVID-19 concerns
[32]	2020	Giavoli C	Italy	Treatment of growth hormone (GH) deficiency	Telephonic survey	208	April 2020	3.4% of patients missed injections due to problems related to drug supply
[42]	2020	Khabbazi A	Azarbaijan	Disease-modifying antirheumatic drugs in patients with autoimmune inflammatory rheumatic diseases	Telephonic survey	858	July 2020	4.0% of patients was non-adherent because of fear of the immunosuppressive effect of medications, 1.9% for symptoms suggestive of COVID-19
[40]	2020	Pineda-Sic RA	Mexico	Disease-modifying antirheumatic drugs in patients with autoimmune inflammatory rheumatic diseases	Web-based questionnaire	345	May 2020	5.6% of patients suspend medications due to lack of availability, and 2.3% for fear of the immunosuppressive effect of medications
[48]	2021	Akour A	Jordan	Chronic drug treatment	Web-based questionnaire	431	May–August 2020	22.7% of patients stopped or decreased medication intake during the COVID-19 lockdown due to an inability to access drugs at clinics
[30]	2021	Awwad MA	Egypt	Anti-glaucoma drugs	Retrospective study on medical records	4326	March 2020–February 2021 vs. March 2019–February 2020	0.8% patients were non-compliant because of lockdown and transportation difficulties
[36]	2021	Barnes A	Australia	Medications for inflammatory bowel disease	Web-based questionnaire	262	May–July 2020	5% of patients chose to stop, reduce dosage, or omit medications as a direct response to concerns about the COVID-19 pandemic
[28]	2021	Cheung CY	Hong Kong	Immunosuppressive medication in kidney transplant recipients	Survey	210	May–September 2020	None of the patients stopped taking immunosuppressive medications unless it was specifically indicated by their healthcare provider
[44]	2021	Costantino F	France	Medications for chronic inflammatory rheumatic diseases	Survey	655	April–May 2020	25.2% of patients suspended or decreased the dosage of one of their drugs due to fear of contagion, 5.6% for symptoms suggestive of infection
[34]	2021	Dorfman L	Israel	Medications for inflammatory bowel disease in paediatric patients	Telephonic survey	244	May–July 2020	2.9% changed or discontinued their medications due to COVID-19
[41]	2021	Fragoulis GE	Greece	Disease-modifying antirheumatic drugs in patients with autoimmune inflammatory rheumatic diseases	Telephonic survey	500	April 2020	2.2% of patients discontinued treatment due to fear of immunosuppression, 3.8% because of lack of resources/drug shortage
[38]	2021	Iborra I	Spain	Immunosuppressants in patients with inflammatory bowel disease	Telephonic survey	234	March–April 2020	10% of patients intentionally postponed at least one scheduled infusion
[45]	2021	Kulhas Celik I	Turkey	Subcutaneous immunotherapy in paediatric patients	Survey	78	May–September 2020	20.5% of patients discontinued therapy for fear of COVID-19 transmission
[39]	2021	López-Medina C	Spain	Anti-rheumatic medications	Web-based questionnaire	644	April–May 2020	6.7% of patients stopped their treatment because they were afraid to develop COVID-19
[47]	2021	Subathra GN	India	Antiglaucoma eyedrops	Telephonic survey	363	April–July 2020	31.4% of patients interrupted treatment or missed doses for non-availability of medicines
[35]	2021	Tilotta G	Italy	Biological therapy in patients with psoriasis, atopic dermatitis, and hidradenitis suppurativa	Retrospective study on medical records	456	March–September 2020	0.4% of patients interrupted treatment for fear of contagion
[43]	2021	Zhang Y	United States	Disease-modifying therapies in patients with multiple sclerosis	Web-based questionnaire	529	April 2020	6.4% stopped or postponed their therapy because of the COVID-19 outbreak
[37]	2022	Caso VM	Italy	Patients frequently undertaking PCSK9i	Telephonic survey	130	March–May 2020	8.5% temporarily interrupted PCSK9i therapy, mostly because of a failure in drug’s prescription due to temporary interruption of the non-urgent outpatient visits and a failure in the drug’s withdrawal due to patients’ fear of becoming infected by leaving the house or taking public transport during COVID-19
[29]	2022	Dorfman L	Israel	Immunosuppressive therapy in paediatric liver transplant patients	Web-based or telephonic survey	76	July–September 2020	none of the patients changed or discontinued their medications due to COVID-19
[49]	2022	Kartal SP	Turkey	Immunosuppressive therapy in patients with psoriasis	Survey	1827	March–July 2020	12.4% interrupted treatment because unable to come to follow-up; 8.2% interrupted treatment for concern about COVID-19
[50]	2022	Konak HE	China	Intravenous immunosuppressive therapy in chronic inflammatory rheumatic diseases	Telephonic survey	181	March 2020–September 2021	14% of patients have postponed at least one dose of their treatment because of fear of COVID-19 disease, 8% for SARS-CoV-2 positivity, and 4% for COVID-19 vaccine.
[46]	2022	Oguz Topal I	Turkey	Systemic therapy in patients with psoriasis	Survey	342	May–August 2021	19.2% of patients discontinued medications due to the inability to go to the hospital, 16.6% for concern about the COVID-19 infection, 7.3% for inability to reach the doctor, 7.3% for inability to have access to the medication, 5.8% for SARS-CoV-2 positivity, 3.8% for COVID−19 vaccine
[33]	2022	Principe R	Italy	Chronic respiratory drugs	Survey	284	June–September 2020	2.8% of patients reported interruption due to expired treatment plan

## 4. Discussion

As the COVID-19 pandemic spread at the beginning of 2020, many countries had to take drastic decisions to protect citizens’ health and safety, such as lockdowns and restrictions on people’s movement and the mobilization of health personnel to the frontline of the COVID-19 infection. In addition, the risk of being infected at hospitals has forced most patients to avoid their health facilities. This may have had major consequences for patients with chronic diseases, requiring follow-up visits, and prescription refills [51]. This systematic review of the literature concerning the impact of the COVID-19 pandemic on adherence to treatment of chronic conditions showed that some chronic therapies were interrupted or affected by reduced adherence in the pandemic period compared with previous years, and that fear of contagion, difficulty in reaching physicians or healthcare facilities, and unavailability of medication were often reported as reasons for treatment discontinuation or modification. In other cases, adherence was preserved during the pandemic.

Overall, the results of surveys, as well as analyses of prescribing trends, depict two distinct behaviours. In some cases, stockpiling was observed at the beginning of the lockdown, probably induced by patients’ fear of running out of medication; this tendency was described for treatments of epilepsy [52] and chronic cardiovascular diseases [53,54,55], and seems to have somewhat preserved adherence in the following months. Some authors have suggested a potential downside to this behaviour, associating the tendency to stockpiling drugs believed to be potentially effective against COVID-19 (often in the absence of adequate evidence-based support) with shortage episodes [56,57]. In other cases, for therapies requiring specific healthcare for their administration, as in the case of parenteral therapies, a delay in scheduled administration was observed. For example, an evaluation of the filled prescription trends for parenteral osteoporosis therapy in Austria [58] showed a continuous increase of prescriptions over the last 2 years, with a remarkable decrease of 22–23% only during the first COVID-19 lockdown in March and April 2020. Even though a subsequent higher number of prescriptions suggests that many patients have received their missed dose later on, this delay could result in an increase the risk for rebound-associated vertebral fractures.

Our review also reveals significant differences, attributable to the type of drug, care setting, and geographical context.

Results from a comprehensive analysis on a dataset of 9.4 billion US prescription drug claims [59] showed that the likelihood of discontinuing therapy was differently modified during the pandemic, depending on the type of drug, being higher for hormonal contraceptive, ADHD stimulant treatments, or antidepressant, and lower for immunosuppressant treatment or opioid addiction therapy. The authors suggested that the drugs less prone to discontinuation are those requiring to be more closely managed by physicians for their administration or monitoring. However, according to a systematic review addressing the frequency and reasons for the disruption of care for inflammatory bowel disease patients [60], the pooled rate of adherence failure with this therapy was 10.12 (CI, 7.12–14.18) per 100 patients, mainly driven by concerns regarding safety amongst both clinicians and patients [61]. In fact, the reduction in adherence reported in some studies [20,21] seems to be mostly related to the fact that patients with rheumatic diseases believed that the immunosuppression obtained with their treatment increased their risk of being infected with COVID-19 or worsen the severity of the disease, and that stopping treatment might reduce the risk [40,42,62,63]. This issue was promptly addressed through specific recommendations and national guidelines [50,63,64], which contributed to mitigate the problem, with small percentages of subjects who discontinued treatment for this reason, or with cases in which treatment was just postponed [41].

Studies on therapies for chronic respiratory diseases, such as asthma or COPD, described improvements [15], worsening [17], or insignificant changes [16] in adherence. In US, Kaye et al. [15] showed that patients enrolled in a digital self-management platform to manage their asthma and COPD maintained higher controller medication adherence throughout 2020. Authors pointed out that this positive trend could be a result of patient concern about controlling their primary respiratory illness during pandemic, but also that the source of data (electronic medication monitors data) may have resulted in a selection bias, leading to the inclusion of patients more motivated. Conversely, the cohort analysis by Ramey et al. on older patients with asthma reported a significantly decreased in adherence. The latter also highlighted that those with a 90-day supply were more likely to be highly adherent to their controller medications, suggesting that disruptions in access during pandemic may have played a key role in reducing adherence to therapy. Concordantly, in the survey by Principe et al. [33] in Italy, the most frequent reason for an interruption or reduction of therapy was the non-renewal of the treatment plan by specialists.

This aspect also emerges from the studies reporting reductions in adherence to therapies that must be administered by experienced personnel at healthcare facilities. The missed infusion of biologics in patients with inflammatory bowel disease [18] or denosumab in patients with osteoporosis [19] is indicative of patients’ inability to reach facilities or of the decision not to go to the clinic due to their fear of infection [38,44,49,50]. For other therapies where the patient was not required to attend the clinic, especially in cases of renewal of an established therapy or minor changes, continuity of treatment was in some cases ensured through the use of telemedicine [24]. The literature reports that diagnoses and treatments made via telephone or other electronic channels increased significantly during the pandemic [65]. The integration of electronic tools in healthcare, strongly and necessarily accelerated by the pandemic, is a positive development that can contribute to the management of chronic diseases even after the pandemic emergency [66,67].

Another issue reported by patients to justify non-adherence to therapy was a drug shortage [40,47,48,68]. This problem has been also described in US [69] and Europe [70], but was certainly much more relevant in low- and middle-income countries (LMICs) [71]. A differential impact of the pandemic in different geographical contexts should also be considered, in terms of infection rates, hospitalizations, and mortality [72]. Healthcare systems in LMICs have been particularly strained by the effect the pandemic has had on already weak health system. The socio-economic gap, together with poor quality access to healthcare of LMICs, became even more evident during COVID-19 period. For patients who even prior to the pandemic could not afford prescription refills and healthy lifestyle adjustments, a deterioration of their condition as a result of poor health accessibility has been reported [73,74]. The global shut down has led to fewer pharmaceutical imports and most pharmaceutical manufacturing firms have shifted their focus to the production of medicines and medical equipment targeted at the fight against COVID-19 in their nations. It will be important to assess the impact of all these factors and the resulting deterioration of the management of patients with chronic diseases on morbidity and mortality in these populations in the medium and long term, in addition to the direct effects of the pandemic.

### 4.1. Strengths and Limitations

The findings of the review are limited by the high variability of methodological approach and by the wide range of rates of adherence found in the scientific literature, this being attributable to the different measures and definitions of adherence used. However, it has to be acknowledged that, in the comparison of adherence during pandemic period vs. pre-pandemic period (primary outcome), all the 12 included studies applied objective measures for adherence assessment, and our result of interest concerned the comparison of the same measure over two time periods, ensuring the robustness of the evidence. The variability was certainly greater for studies assessing our secondary outcome; for this reason, it is not possible to give a quantitative interpretation of the estimates, but only to derive a general picture of the trend of patients’ behavior. Another limitation is that the assessment of the quality of each article using critical reading sheets is open to a degree of subjective interpretation, although we have attempted to compensate for this to some extent by 2 different researchers reviewing each article independently. Finally, the retrieved studies were mostly related to the early period of the pandemic outbreak, and it was therefore not possible to verify a medium-term effect of the introduction of anti-COVID-19 vaccines on patients’ attitudes towards their treatment. Indeed, as an effective protective tool, vaccines may have mitigated the patients’ concerns and increase their adherence to ongoing treatments. Studies evaluating the years 2021 and 2022 are needed to further explore this aspect.

Beyond these limitations, to the authors’ knowledge this is the first study to have systematically searched and analyzed the evidence available in the literature on the impact of the COVID-19 pandemic on medication adherence. The findings provide information to better understand one of the secondary consequences of the pandemic and to guide possible interventions for improvement.

### 4.2. Perspective

Adherence to therapy is a prerequisite for optimising the efficacy of pharmacological treatments and avoiding adverse consequences of worsening or flare-ups of diseases. While the public health effects of the worsening of chronic disease management reported in some settings and in some patient groups need to be evaluated and monitored over time, some positive strategies triggered by the pandemic should be acknowledged.

As already mentioned, telemedicine has gained a primary role during pandemic. Telemedicine has expanded exponentially, supporting access to essential healthcare services and health information, and allowing people with mild symptoms to receive medical consultations from their homes, avoiding risk of infection and reserving physical capacity in healthcare units for critical cases and people with serious health conditions. During the COVID-19 pandemic, patients have found telemedicine a beneficial tool for consulting healthcare providers, with a high level of satisfaction [75]. The literature describes some virtuous examples of the application of telemedicine, which have minimised treatment discontinuities in patients [76]. As an example, in Italy, the success of the use of teledermatology for therapeutic continuity in patients with psoriasis guaranteed patient’s drug accessibility, leading to high therapeutic adherence [35]. Nevertheless, scaling up telemedicine requires high-level political will and support. New investments to create digital platforms and applications, improve access to virtual mental health supports, and expand capacity to deliver healthcare virtually should be included in the health policies of the coming years [77]. On the other hand, potential barriers of implementing e-health tools, such as low digital literacy, low-income, older age, or limited broadband infrastructure, should be taken into consideration.

Another aspect that deserves consideration is the expanded role of community pharmacists. In many countries, the community pharmacist is now in charge of some of the tasks usually covered by doctors, so that doctors are allowed to spend their time more effectively on most complex cases, minimising the number of medical consultations. Over time, the figure of the pharmacist evolved from a ‘drug dispenser’ towards being services-based and patient-centered, with services offered by the pharmacists gradually expanded, including simple medical services, such as measuring blood pressure or vaccines administration, patient education and counselling, and adherence promotion [78]. This process, which had already started in some countries a few years ago, was also greatly and effectively accelerated during the pandemic [79]. Community pharmacists’ roles and responsibilities during the COVID-19 emergency suggest that they are able to play an important role not only in the management of emerging infectious diseases, but also in preserving continuity of care for people with chronic diseases [80].

## 5. Conclusions

Many therapies for chronic conditions were interrupted or affected by reduced adherence in the pandemic period, with some heterogeneity across different settings. The reasons for failure to adhere were a combination of social restrictions and patient-related factors (fear of infection). However, the data also demonstrate that optimal adherence was possible even in wake of ongoing disruptions due to the pandemic. The increasing use of telemedicine, as well as the greater involvement of community pharmacist in the management of chronic patients, could be successful strategies for increasing adherence even after the pandemic. To date, three years after the outbreak of COVID-19 emergency, although the situation is stabilizing, it remains of interest to understand how the observed effects of the pandemic have impacted patients’ attitudes. This evaluation will require future studies.

## Figures and Tables

**Figure 1 ijerph-20-03825-f001:**
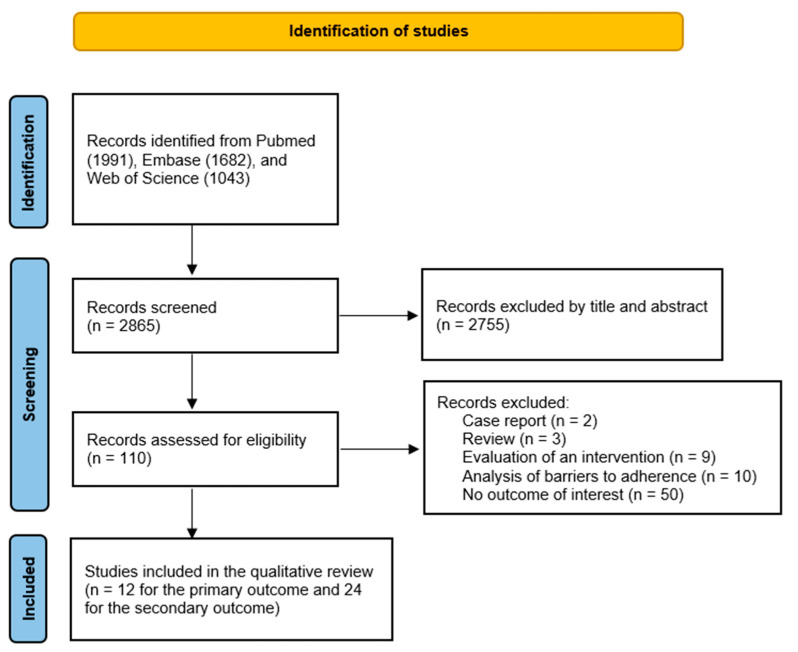
PRISMA flow diagram.

## Data Availability

All the data in the systematic review are from published literature.

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
