# Peer review of "Impact of COVID-19 Pandemic on Adherence to Chronic Therapies: A Systematic Review"

_ijerph, 2023, doi:10.3390/ijerph20053825_

Round 1
Reviewer 1 Report
Overall, I believe this a very good study. The author’s work stems from an interesting premise and provides a good analysis of medication adherence trends during the COVID-19 pandemic.
Regarding the study in general, I would like to simply note that some minor English revision is needed, for example:
“Beyond (to) the direct impact on morbidity and mortality”
I have no other general comments; please find my comments specific top each section below:
Abstract:
The abstract is clear, well-written and provides a good overview of the work.
Introduction:
This section provides a good background for the manuscript. However, it is written as if we are currently in the midst of the COVID-19 pandemic. While COVID-19 is still a global issue and will probably be for years to come, we are now in a transition period where society and institutions are readapting and taking measures to return to normalcy. I would suggest the authors update the introduction with this in mind.
Materials and Methods:
The study’s methodology is sound, and the authors have rightly followed PRISMA guidelines. However, I have a few notes:
- The search query development should be explained. Indeed, questions such as: How as the search query determined? Was it based on previous work, or expert opinions? Was there a step for calibration? should be answered.
- When the authors write that “An example of the complete search strategy is provided below.”, is the implication that there were other search queries that are not detailed in the manuscript? Using the word “example” makes it seem so. Please clarify.
- In lines 91-92 the specification of primary outcome and secondary outcome is unnecessary and adds confusion to the text, as it may induce the reader in thinking that the authors are referring to the included studies’ outcomes instead of the present study’s outcomes. I would advise the authors to remove this. I understand that the authors structured their work around this distinction, and that too is a choice I do not quite understand. Indeed, I would ask them to explore this choice in the methodology and explain why this grouping of studies, which may be a source of bias, was done.
Results:
The results section is very well written and provides a clear, detailed description of the study’s findings.
Discussion:
The discussion section is extremely well-researched and the authors provide rich, detailed and valuable insights into not only adherence as affected by COVID, but medication adherence in general. I would like to congratulate them for their work. My only suggestion is that, as it was in the Introduction, the Discussion at times is written as if we are still going through the pandemic. While this is not necessarily something that needs correction, the authors may want to revise that.
Author Response
Overall, I believe this a very good study. The author’s work stems from an interesting premise and provides a good analysis of medication adherence trends during the COVID-19 pandemic.
We really appreciate the reviewer's positive feedback and we offer the following answers to his/her remarks.
1) Regarding the study in general, I would like to simply note that some minor English revision is needed, for example:
“Beyond (to) the direct impact on morbidity and mortality”
Thank you for the comment. We have carefully revised all the manuscript checking for grammatical errors and misspelling.
2) Abstract:
The abstract is clear, well-written and provides a good overview of the work.
We would like to thank the Reviewer for the comment.
3) Introduction:
This section provides a good background for the manuscript. However, it is written as if we are currently in the midst of the COVID-19 pandemic. While COVID-19 is still a global issue and will probably be for years to come, we are now in a transition period where society and institutions are readapting and taking measures to return to normalcy. I would suggest the authors update the introduction with this in mind.
Thank you for your comment. We have revised the manuscript accordingly.
4) Materials and Methods:
The study’s methodology is sound, and the authors have rightly followed PRISMA guidelines. However, I have a few notes:
- The search query development should be explained. Indeed, questions such as: How as the search query determined? Was it based on previous work, or expert opinions? Was there a step for calibration? should be answered.
Thank you for your comment; we offer the following answer. No previous work was used as reference, given the relative novelty of the topic. We tried to keep the search parameters as broad as possible, to avoid missing potentially interesting articles. The choice of words for the search focused on our scientific question, combining key terms related to therapy adherence and the investigated context of COVID-19 pandemic. We added more details related to this aspect in the methods.
- When the authors write that “An example of the complete search strategy is provided below.”, is the implication that there were other search queries that are not detailed in the manuscript? Using the word “example” makes it seem so. Please clarify.
Thank you for your comment; we offer the following answer. The same combination of terms has been adapted in different format to be applied in PubMed, Embase, and Web of Science databases. For the sake of completeness, we have reproduced all the search strings in the Supplementary Material.
- In lines 91-92 the specification of primary outcome and secondary outcome is unnecessary and adds confusion to the text, as it may induce the reader in thinking that the authors are referring to the included studies’ outcomes instead of the present study’s outcomes. I would advise the authors to remove this. I understand that the authors structured their work around this distinction, and that too is a choice I do not quite understand. Indeed, I would ask them to explore this choice in the methodology and explain why this grouping of studies, which may be a source of bias, was done.
The distinction between primary and secondary outcomes stems from the research question itself. The assessment of the change in adherence due to the pandemic must necessarily be based on a pre-post comparison, and these types of evaluations have seldom analysed the reasons for any change in adherence from the patients’ perspective. This second evidence derives primarily from surveys and questionnaires. The distinction was therefore necessary given the different nature of the study approaches. As rightly noted by the reviewer, this aspect is crucial in the design of our analysis. We therefore decided not to remove the text indicated, but to better specify the rationale for the choice of our outcomes in the methods section.
5) Results:
The results section is very well written and provides a clear, detailed description of the study’s findings.
We would like to thank the Reviewer for the feedback.
6) Discussion:
The discussion section is extremely well-researched and the authors provide rich, detailed and valuable insights into not only adherence as affected by COVID, but medication adherence in general. I would like to congratulate them for their work. My only suggestion is that, as it was in the Introduction, the Discussion at times is written as if we are still going through the pandemic. While this is not necessarily something that needs correction, the authors may want to revise that.
Thank you for your comment. We have carefully revised the manuscript accordingly.
Reviewer 2 Report
This is a well-designed and well-performed systematic review of literature on impact of Cvidd-19 pandemic on adherence to chronic therapies, of high interest to the general audience.
Only minor modification are advisable, such as:
1. Figure 1 would be more informative if the reasons for exclusion of 2k records are provided
2. Table 1 would be more informative if the specific method of calculation of adherence was provided in the ‘results’ column, such as e.g. PDC
3. Table 2 is of minor importance for general audience please consider moving it to the online appendix
4. A WHO 2020 report, and 2 publications from ENABLE group (Agh, 2021; Kardas, 2021) assessing the impact of Covid-19 on adherence are worth adding to the introduction, due to their international scope.
Besides, a minor typo is to be corrected in line 62 (‘SARS-CoV-2infections’), and the values in Table 2 should be put in capital letters, just as the legend shows.
Author Response
This is a well-designed and well-performed systematic review of literature on impact of Cvidd-19 pandemic on adherence to chronic therapies, of high interest to the general audience.
We would like to thank the Reviewer for the positive feedback and suggestions that we have addressed in the revised manuscript. Please find below a point-by-point response to the comments.
1) Figure 1 would be more informative if the reasons for exclusion of 2k records are provided
Thank you for your comment; we offer the following answer. We modified the flow diagram (Figure 1) accordingly.
2) Table 1 would be more informative if the specific method of calculation of adherence was provided in the ‘results’ column, such as e.g. PDC
Thank you for your comment; we offer the following answer. To this end we moved the measure of adherence column next to the results column.
3) Table 2 is of minor importance for general audience please consider moving it to the online appendix
Thank you for your comment; we offer the following answer We moved this table in the supplementary material (Supplementary Table 1).
4) A WHO 2020 report, and 2 publications from ENABLE group (Agh, 2021; Kardas, 2021) assessing the impact of Covid-19 on adherence are worth adding to the introduction, due to their international scope.
Thank you for the suggestion. We have now included these references.
5) Besides, a minor typo is to be corrected in line 62 (‘SARS-CoV-2infections’), and the values in Table 2 should be put in capital letters, just as the legend shows.
Thank you for your comment; we offer the following answer. We have carefully revised all the manuscript checking for grammatical errors and misspelling. We also modified values in Table 2 (now Supplementary Table 1).